# Do dogs eavesdrop on human interactions in a helping situation?

**Hoi-Lam Jim** 🄳 *, **Sarah Marshall-Pescini, Friederike Range***

Wolf Science Center, Domestication Lab, Konrad Lorenz Institute of Ethology, Department of Interdisciplinary Life Sciences, University of Veterinary Medicine Vienna, Vienna, Austria

\* hoi-lam.jim@vetmeduni.ac.at (HLJ); friederike.range@vetmeduni.ac.at (FR)

**Data Availability Statement:** All relevant data are within the manuscript and its Supporting information files.

**Funding:** HLJ and FR were funded by the Austrian Science Fund (FWF): W1262-B29 (www.fwf.ac.at). The funders had no role in study design, data

## Abstract

Eavesdropping is the acquisition of information by observing third-party interactions. Considering dogs' (*Canis lupus familiaris*) dependence on humans, it would be beneficial for them to eavesdrop on human interactions to choose an appropriate partner to associate with. Previous studies have found that dogs preferred a human who acted generously or cooperatively towards another human over one who acted selfishly or non-cooperatively, however they often did not control for potential location biases. This study controlled for local enhancement and investigated whether dogs derive and act on information about unfamiliar humans through reputation-like inferences by observing third-party interactions. 42 dogs participated in the experiment, which consisted of an observation phase and a test phase. In the observation phase, the animals observed a human with a box of food ask for help to open it from two people—one was helpful and the other was not. The test phase consisted of the impossible task and a choice test. Half of the sample was tested in the experimental condition and the other half was tested in the side control condition, where the two people swapped positions before the test phase. The results of the impossible task showed that dogs only looked at the helpful person first when the people stayed on the same side as they did in the observation phase. In the choice test, dogs chose at random, regardless of whether the people stayed on the same side or swapped positions. Our findings provide tentative support for a local enhancement interpretation of eavesdropping.

## Introduction

Reputation represents a set of beliefs, perceptions or evaluative judgments about an individual's typical behaviour based on knowledge of that individual's past behaviour [1]. It is a key component of social interactions in group-living animals and may have played an important role in the evolution of cooperation [2], as an individual with a reputation of being cooperative may be more likely to gain access to valuable resources and partners than an individual with a reputation of being non-cooperative, who may be excluded from social interactions instead [3]. Thus, having a good reputation can contribute to survival [4].

Animals may form a reputation of another individual by directly interacting with it or by observing it interacting with a third-party, i.e., eavesdropping [5]. Eavesdropping may be

collection and analysis, decision to publish, or preparation of the manuscript.

**Competing interests:** The authors have declared that no competing interests exist.

more cognitively demanding, as it requires individuals to remember and recognize behaviours in third-party interactions, but it serves a vital function in allowing animals to predict the behaviour of others without the potential costs of direct experiences [6]. Despite this perceived importance, only a few studies have shown that non-human animals eavesdrop. A study on client-cleaner reef fish *Labroides dimidiatus* interactions showed that eavesdropping clients spent more time next to 'cooperative' cleaners than ones they had no knowledge of their cooperativeness [7]. There are several studies on non-human apes, which involved animals observing a 'nice' person giving food to a human beggar and a 'nasty' person who refused to give food, and these have shown mixed results. Russell, Call and Dunbar [1] tested chimpanzees (*Pan troglodytes*), bonobos (*Pan paniscus*), gorillas (*Gorilla gorilla*) and orangutans (*Pongo pygmaeus*) and found that chimpanzees spent more time near the nice than the nasty person, whereas the results for the other apes were not significant. Herrmann, Keupp, Hare, Vaish and Tomasello [8] tested chimpanzees, bonobos and orangutans and found that chimpanzees and orangutans preferred to approach the nice experimenter rather than the nasty experimenter, however bonobos did not show a preference for either experimenter. Subiaul, Vonk, Okamoto-Barth and Barth [6] also tested chimpanzees in the same setup and did not find any significant results. They conducted a follow-up experiment where a conspecific was the recipient of the interaction and found that chimpanzees preferred the nice person over the nasty person. However, it could not be ruled out that the chimpanzees learned to attend to the nice experimenter rather than forming a reputation of "generous" in this study.

A problem with studying eavesdropping in non-human apes is that the situation is highly artificial, as they do not typically rely on humans for social information. Therefore, studying animals that live and interact with humans, i.e., domesticated species, would be more ecologically valid. A recent study by Leete, Vonk, Oriani, Eaten and Lieb [9] showed that domestic cats (*Felis silvestris catus*) did not attribute reputations to humans through direct or indirect interactions between humans and conspecifics. A possible explanation for this result is that cats are typically solitary animals and do not rely on cooperation with others for survival. Conversely, dogs (*Canis lupus familiaris*) descend from wolves (*Canis lupus*), which live in family groups and engage in group hunting and alloparenting [10], thus they are highly social and cooperative. Furthermore, dogs were domesticated by humans—they can cooperate with humans and rely on them for valuable resources [11], such as food and shelter. Thus, it would be beneficial for dogs to observe humans interacting with each other or with other dogs and gain information from such an exchange to select the most appropriate person with whom to associate and whom to avoid.

Rooney and Bradshaw [12] found that dogs preferred to approach a person who won a tug-of-war game with another dog over a person who lost the game. In contrast, Nitzschner, Melis, Kaminski and Tomasello [13] found that dogs did not have a preference for a nice compared to an 'ignoring' experimenter after observing them interact with another dog. However, they did find that subjects behaved differently during the nice and ignoring demonstrations, which suggests that they were attentive to the different types of dog-human interactions but the experimenters' behaviour towards the demonstrator dog may not have been relevant enough for the observer dog to form reputations of them.

Eavesdropping studies that involved a food-sharing situation have found that dogs preferred to interact with a generous person over a selfish one after indirect experience with them [14,15]. However, the dogs' behaviour in these studies could be explained by simpler mechanisms, such as choosing to approach the experimenter based on where they saw food being exchanged, i.e., local enhancement. This was not controlled for in Marshall-Pescini, Passalacqua, Ferrario, Valsecchi and Prato-Previde's [15] study. Therefore, Freidin, Putrino, D'Orazio and Bentosela [11] and Nitzschner, Kaminski, Melis and Tomasello [16] tested this hypothesis

by swapping the generous and selfish experimenters' positions before the dogs could choose which experimenter to approach. Their results showed that dogs chose the experimenter on the side where the food interaction had previously happened [16] or chose at random [11], suggesting that results from previous studies could be largely explained by local enhancement. However, Kundey et al. [14] controlled for local enhancement and found that dogs continued to demonstrate reputation-like inferences and chose the generous experimenter after they swapped positions.

Chijiiwa, Kuroshima, Hori, Anderson and Fujita [17] suggested that dogs may have simply preferred the generous donor in the previous studies because that person was associated with food, i.e., stimulus enhancement. Therefore, they tested whether dogs could distinguish two humans in a helping situation. One experimenter helped the dog's owner open a container with a neutral object inside and the other experimenter did not help. Importantly, the experimenters never touched the object involved in the interaction. They found that dogs avoided the experimenter who behaved non-cooperatively with their owner. However, the experimenters did not swap positions before the dogs could choose whom to approach, thus dogs may have relied on local enhancement to make their choice in this study.

Chijiiwa et al. [17] is the only study that has tested eavesdropping in dogs in a helping situation, and every study to date on this topic has measured whether dogs attributed reputations to humans based on the dog's first approach or time spent in close proximity to each experimenter. Another way to measure dogs' preference for a human in a helping situation is the 'impossible task' [18]. This a well-established paradigm where the dog is presented with an apparatus containing food that cannot be accessed. This usually elicits looking behaviour at a human, which has been interpreted as a communicative act to request help from humans [19–26]. When two people are present during this task, it can be used to test whether dogs can discriminate between people—for example, Marshall-Pescini, Passalacqua, Barnard, Valsecchi and Prato-Previde [23] found differences in dogs' looking behaviour towards their owner and the researcher based on their life experiences, whereas Piotti, Spooner, Jim and Kaminski [26] found no significant difference in dogs' looking behaviour between a skilful human and an unskilful one. We used a variation of the impossible task that was similar to Piotti et al.'s [26] first experiment to test whether dogs would look at a helpful person first or for significantly longer over an unhelpful person after observing them acting helpfully or unhelpfully towards another person, possibly to request help to open the box to retrieve the food inside.

The aim of the current study was to clarify whether dogs are indeed capable of eavesdropping or if their success can be explained by a low-level mechanism, such as local enhancement. We also improved previous methodologies used to test eavesdropping: first, we combined elements of the food-sharing situation [14–16] and the helping situation [17]. In our study, dogs observed a human beggar with a box of food ask two people to open it—one experimenter acted helpfully and the other did not. Unlike previous studies, the beggar stood equidistant to the two experimenters when she ate the food to control for stimulus enhancement. Moreover, to control for local enhancement, for half of our study sample, the experimenters swapped positions before the test. Second, like previous studies, we measured dogs' preference considering their choice to approach an experimenter (helpful or unhelpful) after witnessing the third-party interactions. Additionally, we presented the animals with the impossible task after the 'eavesdropping event', and measured their looking behaviour towards the two people, thus presenting a situation to dogs in which they themselves needed help, after witnessing a 'helping event' towards a third party.

We hypothesized that dogs derive and act on information about unfamiliar humans through reputation-like inferences by observing third-party interactions. We predicted that dogs will look first and/or significantly longer at a person who was helpful towards another

person than an unhelpful person in the impossible task and will choose the helpful person in the choice test. The alternative hypothesis is that dogs do not discriminate between a helpful and unhelpful person and simply prefer the side they saw the box being opened during the third-party interaction, i.e., local enhancement. If this is the case, we predicted that dogs would look first and/or significantly longer at the person on the side where they observed the helpful interaction in the impossible task and would choose the person on that side in the choice test.

## Materials and methods

### Ethical statement

Ethical approval for this study was obtained from the 'Ethik und Tierschutzkommission' of the University of Veterinary Medicine Vienna (Protocol number ETK-14/12/2017) and informed consent was obtained from all dog owners. The individuals in this manuscript have given written informed consent (as outlined in PLOS consent form) to publish the videos in the S1 and S2 Videos.

### Subjects

42 pet dogs, 20 males and 22 females, aged between 1 and 13 years old ($M$ = 5.5, $SD$ = 3.3) participated in the study (S1 Table). They were selected from a database of owners who had volunteered to participate in behavioural studies at the Clever Dog Lab and were included on the basis of lack of experience with the experimenters in this study. 25 dogs had participated in other studies. No breeds were excluded.

### Apparatus

There were three identical 150 (L) × 80 (W) × 80 (H) mm sliding-lid boxes with a 15 mm hole in the centre of the lid (Fig 1a). They were made of clear polycarbonate so the dog could see and smell the food. There were three pieces of sausage inside two of the boxes, which were each placed in a hip bag. There was an additional piece of sausage in the front pocket of the two hip bags, which was used in the choice test.

The beggar held the third box, which had five pieces of sausages inside at the beginning of each trial. The beggar's box had two holes on the side for placement in the impossible task. The impossible task apparatus was a 3 m (L) × 15 cm (W) wooden plank with a 152 (L) × 82 (W) × 100 (H) mm clear polycarbonate box with no lid attached at the centre of it (Fig 1b). When the beggar's box was placed inside the clear box in the centre of the impossible task apparatus and two 3 cm nails were slid through the walls of both boxes, the beggar's box could not be displaced or opened, thus the food inside was inaccessible (Fig 1c).

### Experimental setup

The study was conducted in a large empty testing room (7.13 × 6.01 m) with two doors at the Clever Dog Lab. The beggar used one door to enter and exit the room and this was counterbalanced across subjects. The whole room was recorded during the procedure using three video cameras in three corners of the room.

There were three experimenters: the main experimenter, who acted as the beggar and set-up the trials, and two female experimenters who were unfamiliar to the dog. One wore white clothes and the other wore black clothes. Each experimenter wore a hip bag facing backwards and a chest harness with a GoPro Hero 4 attached. In total, six females acted as experimenters in the study and the experimenters' clothes, roles and positions were counterbalanced across

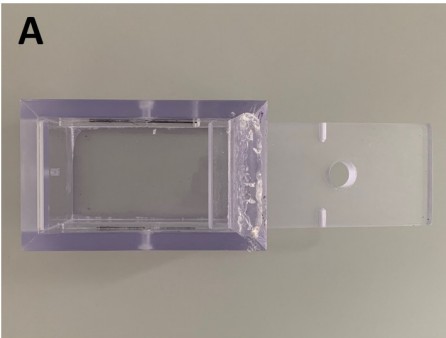
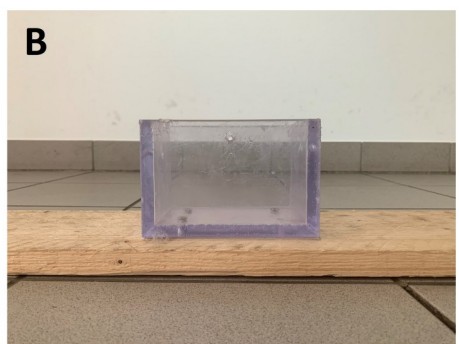
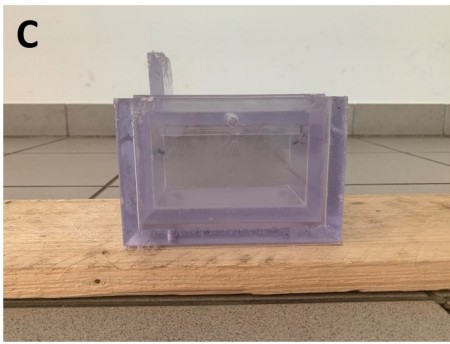

**Fig 1. Photos of the box and impossible task apparatus.** (A) Bird's eye view of the box: there were three identical clear polycarbonate boxes with a hole in the centre of the sliding lid; (B) impossible task apparatus: a wooden plank with a clear polycarbonate box attached at the centre of it; (C) the beggar's box was placed inside the impossible task apparatus and secured by sliding two nails through the walls of both boxes.

subjects. The experimenters used both doors to enter the room and which door they used depended on their position in the trial (i.e., the experimenter on the left used the left door).

Before the trial with the impossible task began, the apparatus was placed in a predetermined position in the centre of the room, 1.5 m away from the dog (Fig 2). During the observation phase, the beggar stood in position B, 3.5 m away from the dog. The two experimenters stood 1.5 m away on either side of the beggar in position A and C. During the test phase, the experimenters walked 2 m forward and stood in position D and E. Two semi-circles with a radius of 1 m from positions D and E facing towards each other were marked by tape on the floor to indicate when the dog was in close proximity to the experimenter.

## Experimental design

Dogs were randomly assigned to two experimental groups: experimental and side control. In the experimental group, the experimenters stayed on the same side of the room in the test phase. In the side control group, the experimenters swapped positions in the test phase to test the local enhancement hypothesis. Half of the sample was in the experimental group and the other half was in the side control group.

## Procedure

Prior to testing, the subject was allowed to explore the room and the impossible task apparatus freely for approximately 5 minutes while the main experimenter explained the procedure to

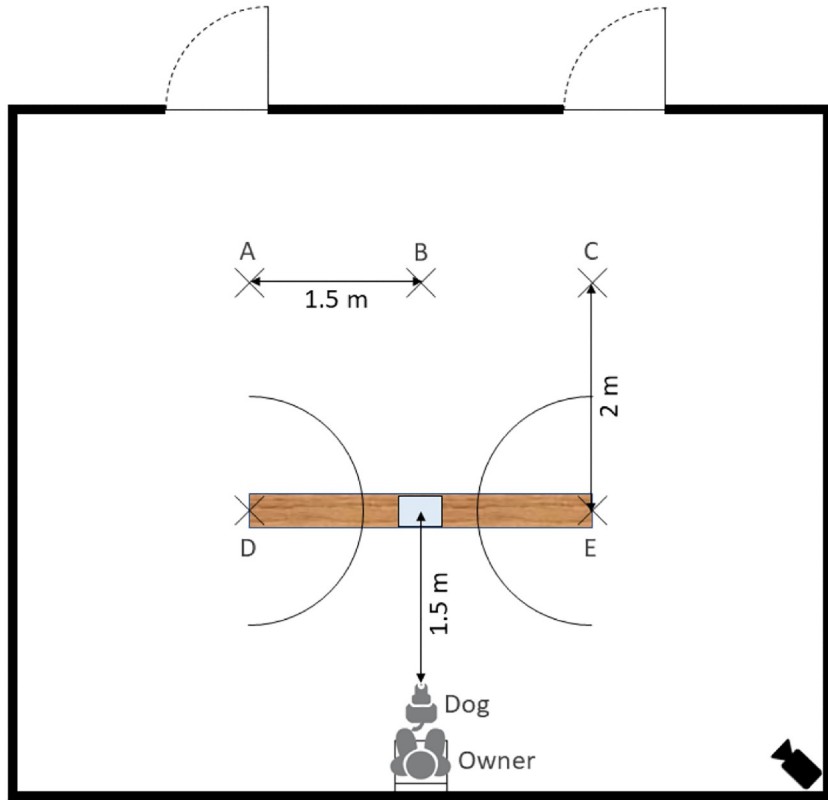

**Fig 2. Schematic depiction of the experimental set-up.** A and C indicate where the experimenters stood and B indicates where the beggar stood in the observation phase. D and E indicate where the experimenters stood in the test phase. The positions were marked by tape on the floor.

the owner. Once the dog was comfortable, the owner sat and held his/her unleashed dog by the collar between his/her legs while the main experimenter placed a piece of sausage in the clear box in the centre of the impossible task apparatus. Then, the owner released the dog and it could eat the food. This was repeated three times so the dog had experience of eating food from the box and so the dog understood that they were free to move when the owner let go of the collar. During the experiment, the owner was blindfolded to prevent him/her from influencing the dog's behaviour and remained seated, holding his/her unleashed dog by the collar between his/her legs.

The experiment consisted of two trials and each trial consisted of two parts: an observation phase and a test phase. In the observation phase, the animals could see the beggar holding a box with food inside and ask for help to open it from the two experimenters. The observation phase was identical for all subjects but the test phase varied according to the group. The test phase consisted of two tests: (1) the impossible task and (2) the choice test. Which of the two tests was conducted first (impossible task or choice test) was counterbalanced across subjects within the experimental groups. After the first trial, the owner and the dog had a short break while the main experimenter set-up the next trial.

**Observation phase.** The beggar entered the room, approached the dog and allowed it to inspect the box. Then, she stood in a predetermined position in the centre of the room (position B in Fig 2) and the experimenters entered the room through both doors and stood in their predetermined positions, 1.5 m away on either side of the beggar (positions A and C in Fig 2).

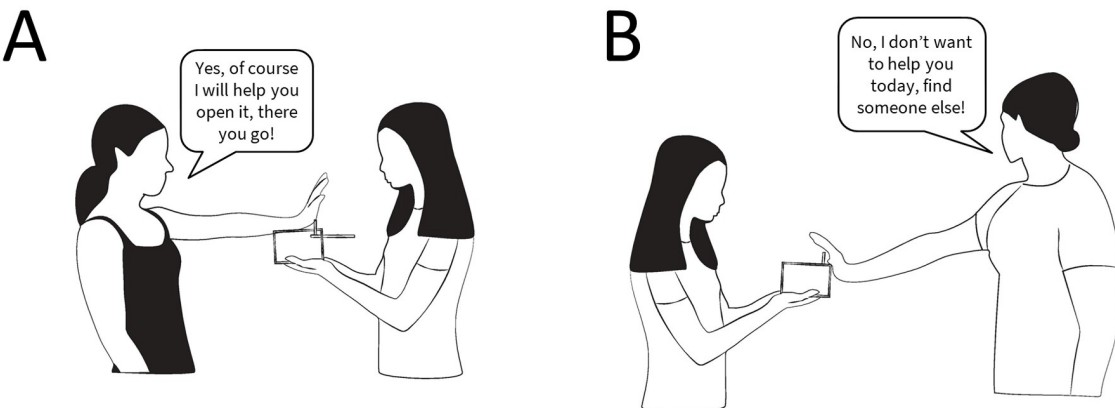

**Fig 3. Illustration of the beggar interacting with (A) the helpful person and (B) the unhelpful person.** By Hoi-Lam Jim and Nadja Kavcik-Graumann.

The beggar approached one of the experimenters and said, "Can you help me?" and the experimenter reacted accordingly to their role: the helpful person said, "Yes, of course I will help you open it, there you go" in a friendly tone and slid the box lid open. In the case of the unhelpful person, she said, "No, I don't want to help you today, find someone else" in a firm voice and pushed the box away whilst turning her head away. Both experimenters performed the same action of extending their arm but the outcome of their action was different—the helpful person opened the box so the food could be accessed and the unhelpful person did not open the box (Fig 3).

After the response, the beggar returned to her position in-between the experimenters and faced the dog. In the instance of having interacted with the helpful person, she took a piece of sausage out of the box, showed it to the dog then put it in her mouth and chewed (i.e., moved her jaw up and down) to make it clear to the dog that she was eating. At the same time, she moved the box towards her chest to close the lid inconspicuously, so that it was closed for the next interaction. In the instance of having interacted with the unhelpful person, the beggar scratched her chin whilst the box remained closed. Thus, the beggar's actions were similar after interacting with both experimenters to ensure the dog did not prefer the helpful person because the beggar performed a particular action after interacting with her (see S1 and S2 Videos).

The beggar moved from one experimenter to the other a total of six times in a semi-random order, never begging more than twice from the same person and spending the same amount of time next to each person. At the end of the observation phase, the experimenters walked 2 m forward and stood in their predetermined positions (positions D and E in Fig 2).

**Test phase.** *(1) Impossible task*. The experimenters stood on each end of the wooden board to prevent it from moving and pressed record on their GoPros. The beggar secured the box into the impossible task apparatus and then took a piece of sausage out of her pocket, showed it to the dog and dropped it into the box, thus leaving three pieces of sausages inside. When the beggar left the room and shut the door, this indicated the start of the test to the owner, who then released the dog. The owner was allowed to give a short prompt if the dog did not move by itself, such as a gentle nudge or saying "ok" to indicate to the dog that they were free to move, but the owner was instructed not to gesture in a specific direction. The owners were blindfolded so they were not aware of the experimenters' positions and could not influence their dog's behaviour. The experimenters stood still, remained silent and kept their

gaze on the box while the dog was free to move about within the room. The test ended after two minutes, then the owner called the dog back and the experimenters exited the room.

*(2) Choice test.* The experimenters twisted their hip bags around to the front, took the box (with three pieces of sausages inside) out of the bag and held it at the level of their navel. They also took the single piece of sausage out of the front pocket of the bag and held it in their hand. When the experimenters were ready, the beggar left the room and the owner released the dog. The experimenters kept their gaze on their box and when the dog approached within 1 m of an experimenter (marked by tape on the floor), she fed it. Then, the other experimenter attracted the dog's attention and also fed it to ensure it did not form a preference for one experimenter, and then the test ended. If the dog did not make a choice within one minute, the test ended and the owner called the dog back. Neither experimenter fed the dog and they exited the room.

## Behaviour analysis

For the impossible task, we coded the footage from the experimenters' GoPros, which was synchronized and merged into one video, and we supplemented it with the wide-angle video footage of the whole room when appropriate (see videos in S1 and S2 Videos). The Solomon Coder (beta 17.03.22, copyright 2017 by András Péter) software was used to code the frequency and duration of behaviour, which was measured from the moment the door closed and concluded after two minutes.

We coded the following behaviours at 0.2 s time resolution: the frequency of (1) 2-way gaze alternations and (2) 3-way gaze alternations and the duration of (3) looking at the box; (4) interacting with the box; (5) looking at the experimenter; (6) interacting with the experimenter; (7) proximity to the experimenter (see Table 1 for definitions). All behaviours towards the helpful and unhelpful experimenter were recorded separately. The dog's first look towards the helpful or unhelpful experimenter after looking at or interacting with the box was also recorded.

For the choice test, the dog's first approach towards the helpful or unhelpful experimenter was recorded and if the dog did not approach either person during the test, it was coded as a 'no-choice' response.

## Statistical analysis

**(1) Impossible task.** One dog (Sixtus) was excluded because he did not look at or interact with the box during the test, so it is plausible that he did not understand the task. To test

**Table 1. Definitions of coded behaviours.**

| Behaviour | Definition |
| --- | --- |
| **(1) 2-way gaze alternation (frequency)** | Unbroken looks between the experimenter and the box or vice versa |
| **(2) 3-way gaze alternation (frequency)** | Unbroken looks between the experimenter and the box and then back to the experimenter or vice versa |
| **(3) Looking at the box (duration)** | The dog orienting its head and/or eyes towards the box |
| **(4) Interacting with the box (duration)** | The dog being in close contact with the box, e.g., sniffing and pawing |
| **(5) Looking at the experimenter (duration)** | The dog orienting its head and/or eyes towards the experimenter |
| **(6) Interacting with the experimenter (duration)** | The dog being in close contact and exhibiting social behaviours towards the experimenter, e.g., sniffing and tail-wagging whilst in close contact |
| **(7) Proximity to the experimenter (duration)** | The dog having two paws within 1 m of the experimenter (marked by tape on the floor) |

whether dogs looked at the helpful experimenter first, we fitted a generalized linear model (GLM) with binomial error structure and logit link function [27]. We estimated model stability by means of dfbeta [28]. The sample size was 39, as two dogs never looked at either experimenter during the test.

We modelled the behaviour of the dogs in terms of the proportion of time they (1) looked at, (2) spent interacting with or (3) were in close proximity to the helpful experimenter. Hence, in the response variable, values above 0.5 indicate a bias towards the helpful experimenter and values below 0.5 indicate a bias towards the unhelpful experimenter. We modelled these responses using GLMs [27] with beta error distribution and logit link function [29], one for each of the three response variables (looking, interacting and proximity). As predictors, the models comprised experimenter position (factor with two levels: same or swap positions) and test order (factor with two levels: choice test or impossible task first) and the interaction between the two. To avoid 'cryptic multiple testing' [30] and keep type *I* error rate at the nominal level of 0.05, we compared each full model with a respective null model lacking experimenter position and its interaction with test order.

We counted the combined number of 2-way and 3-way gaze alternations towards the helpful and unhelpful experimenters and ran a generalized linear mixed model with binomial error structure and logit link function. In R, such a model can be fitted by transforming the response into a two-column matrix with the number of gazes at the helpful and unhelpful experimenters respectively and it needs to be a mixed model to take multiple gazes of the same individual into account [31].

We fitted the models in R (version 3.5.1) [32] using the function betareg of the package betareg (version3.1–2) [33], with the exception of the gaze alternation model, which we fitted using the function glmer of the package lme4 (version 1.1-18-1) [34]. We estimated model stability by excluding individuals one at a time and comparing model estimates derived for these subsets of data with those obtained for the entire dataset (we report results of this stability estimation in the results section). Overdispersion was no issue in any of the models (dispersion parameters for looking: 1.077; interacting: 1.084; proximity: 0.957; gaze alternation: 0.997). The sample size for these models vary because not all dogs showed all behaviours considered (looking: $N = 35$; interacting: $N = 34$; proximity and gaze alternation: $N = 37$).

We also tested whether, overall, the proportion of time individuals showed the three response variables (looking, interacting and proximity) towards the helpful experimenter and the probability of gaze alternations towards the helpful experimenter differed from the chance expectation of 0.5. To this end, we manually dummy coded and centred both factors and fitted a model lacking the interaction between them. In such a model, the estimate for the intercept reveals the deviation from chance across the whole dataset and the *p* value for it tests whether this deviation from chance is significant.

**(2) Choice test.**   To test whether dogs chose the helpful experimenter over the unhelpful experimenter, we fitted a GLM with binomial error structure and logit link function and we estimated model stability by means of dfbeta. The sample size was 35, as seven dogs did not choose either experimenter during the test.

## Results

The GLM conducted to test whether dogs looked at the helpful experimenter first in the impossible task revealed that the proportion of dogs that looked at the helpful experimenter first was significantly above chance when the experimenters stayed in the same position ($p = .033$). However, when the experimenters swapped positions, the proportion of dogs that looked at the helpful experimenter first was not significantly different from chance (Fig 4,

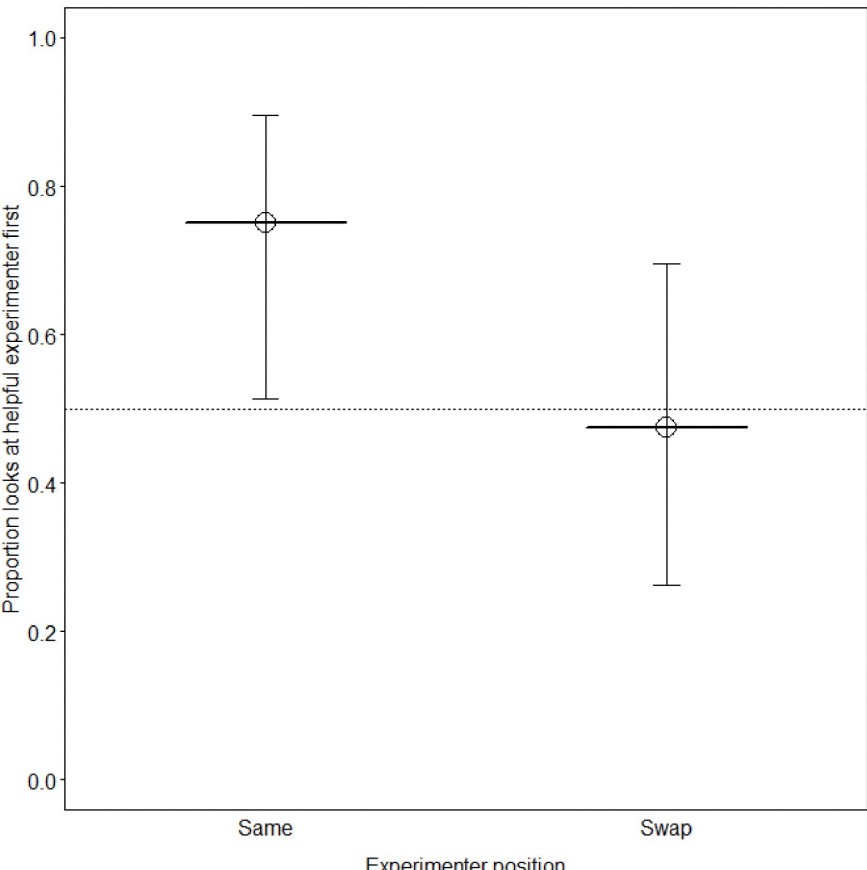

**Fig 4. Proportion of dogs that looked at the helpful experimenter first in the impossible task.** The solid horizontal lines depict the fitted model and the dotted horizontal line depicts chance expectation.

Table 2). We also conducted a GLM to test whether dogs chose the helpful person in the choice test, which revealed no significant difference in the proportion of choices for the helpful experimenter when the experimenters stayed on the same side or swapped positions ($p = .397$) (Fig 5, Table 2).

**Table 2. Results of the full models.**

| Response variable | Term | Estimate | SE | 95% CI | | z | p | Min | Max |
|---|---|---|---|---|---|---|---|---|---|
| | | | | Upper | Lower | | | | |
| **Proportion of dogs that looked at the helpful experimenter first in the impossible task** | Intercept | 1.099 | 0.516 | 0.150 | 2.222 | 2.127 | .033 | 0.896 | 1.191 |
| | Experimenter position[a] | -1.204 | 0.691 | -2.628 | 0.115 | -1.742 | .082 | -1.330 | -1.002 |
| **Proportion of dogs that chose the helpful experimenter in the choice test** | Intercept | 0.357 | 0.493 | -0.600 | 1.370 | 0.724 | .469 | 0.188 | 0.487 |
| | Experimenter position[a] | -0.580 | 0.684 | -1.956 | 0.751 | -0.848 | .397 | -0.711 | -0.411 |

Estimates, standard error, confidence intervals, results of significance tests and minimum and maximum of model estimates derived after excluding individuals one at a time.

[a]Dummy coded: 0 = 'same'; 1 = 'swap'.

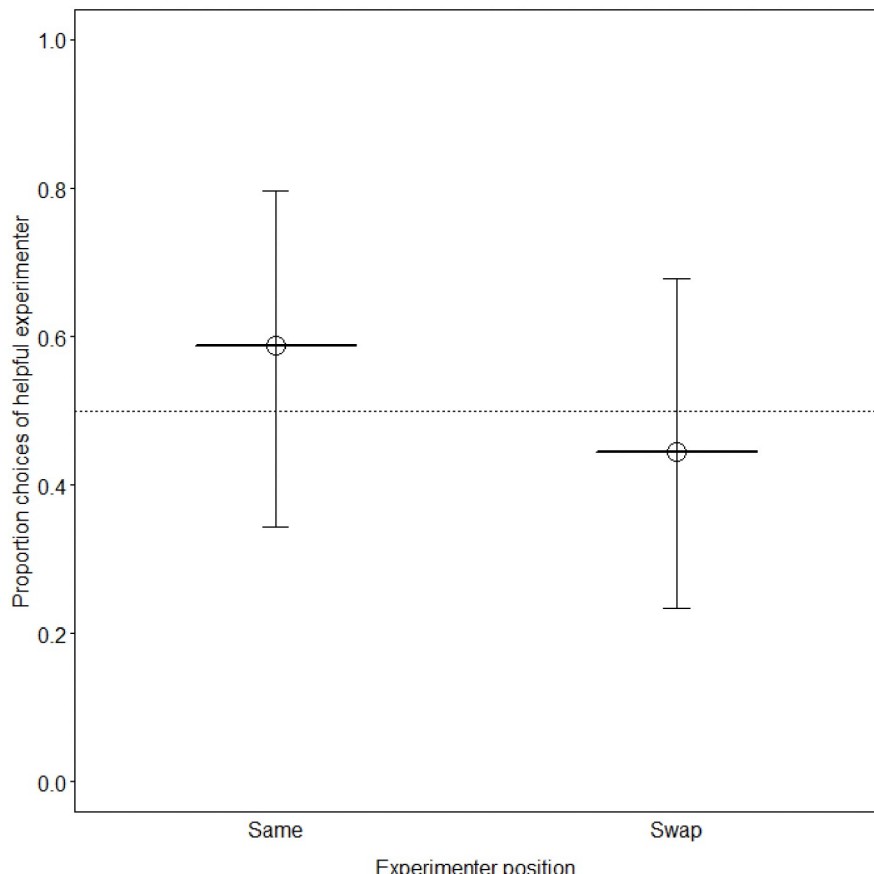

**Fig 5. Proportion of dogs that chose the helpful experimenter in the choice test.** The solid horizontal lines depict the fitted model and the dotted horizontal line depicts chance expectation.

The GLMs conducted to test whether dogs preferred the helpful experimenter over the unhelpful one in the impossible task revealed no significant difference across the four response variables in any of the full-null model comparisons (likelihood ratio tests: looking: $\chi^2$ (2, $N = 35$) = 3.178, $p = .204$; interacting: $\chi^2$ (2, $N = 34$) = 0.388, $p = .824$; proximity: $\chi^2$ (2, $N = 37$) = 1.595, $p = .450$; gaze alternation: $\chi^2$ (2, $N = 37$) = 2.908, $p = .234$) (S2 Table).

After dropping the interaction, which was non-significant in all models, we did not find a main effect of experimenter position or test order in any of the four reduced models, although there was a marginal trend for the effect of experimenter position for looking and gaze alternation (see Table 3). However, since the estimates for experimenter position in these two models pointed in opposite directions (i.e., dogs looked more and gaze alternated less at the helpful experimenter when they swapped positions) and neither of the full-null model comparisons revealed significance, these results should be considered carefully, as they could represent type I errors.

We also conducted tests of the intercept with both factors (experimenter position and test order) centred to test whether, on average, the four response variables towards the helpful experimenter differed from chance expectation. This was non-significant for all four response variables, although there was a weak trend towards gaze alternating more at the helpful experimenter (looking: Estimate ± $SE$ = -0.028 ± 0.201, $z = -0.14$, $p = 0.889$; interacting: Estimate ±

**Table 3. Results of reduced models (lacking the interaction) in the impossible task.**

| Response variable | Term | Estimate | SE | 95% CI | | z | p |
|---|---|---|---|---|---|---|---|
| | | | | Lower | Upper | | |
| **Proportion of time dogs looked at the helpful experimenter** | Intercept | -0.425 | 0.362 | -1.134 | 0.284 | -1.174 | .241 |
| | Experimenter position[a] | 0.687 | 0.411 | -0.119 | 1.492 | 1.671 | .095 |
| | Test order[b] | 0.089 | 0.404 | -0.702 | 0.880 | 0.22 | .825 |
| **Proportion of time dogs interacted with the helpful experimenter** | Intercept | 0.163 | 0.344 | -0.511 | 0.837 | 0.475 | .635 |
| | Experimenter position[a] | 0.198 | 0.389 | -0.564 | 0.959 | 0.509 | .611 |
| | Test order[b] | -0.575 | 0.394 | -1.349 | 0.198 | -1.459 | .145 |
| **Proportion of time dogs spent close to the helpful experimenter** | Intercept | -0.054 | 0.287 | -0.616 | 0.508 | -0.189 | .850 |
| | Experimenter position[a] | -0.391 | 0.318 | -1.014 | 0.233 | -1.228 | .219 |
| | Test order[b] | 0.224 | 0.318 | -0.400 | 0.847 | 0.703 | .482 |
| Response variable | Term | Estimate | SE | 95% CI | | $\chi^2$ | df | p |
| | | | | Upper | Lower | | | |
| **Proportion of time dogs gaze alternated to the helpful experimenter** | Intercept | 0.108 | 0.238 | -0.355 | 0.618 | | | |
| | Experimenter position[a] | -0.484 | 0.283 | -1.073 | 0.070 | 2.908 | 1 | .088 |
| | Test order[b] | 0.469 | 0.285 | -0.055 | 1.054 | 2.727 | 1 | .099 |

Estimates, standard error, confidence intervals and results of significance tests.

[a]Dummy coded: 0 = 'same'; 1 = 'swap'.

[b]Dummy coded: 0 = 'choice test first'; 1 = 'impossible task first'.

$SE$ = -0.065 ± 0.194, $z$ = -0.336, $p$ = 0.737; proximity: Estimate ± $SE$ = -0.123 ± 0.158, $z$ = -0.779, $p$ = 0.436; gaze alternation: Estimate ± $SE$ = 0.361 ± 0.201, $z$ = 1.802, $p$ = 0.071).

## Discussion

Our study investigated whether dogs eavesdrop on human interactions in a helping situation and if so, whether their success can be explained by local enhancement. We found that when dogs needed help to open an apparatus, they looked at a person who had been helpful towards another person first rather than at the unhelpful person. However, this effect was only found when the two people stayed on the same side of the room before the impossible task and there was no significant difference between which experimenter dogs looked at first when they swapped positions. Our results are in line with Freidin et al. [11], who found that dogs preferentially approached a person who had received positive reactions from the beggar when the people stayed on the same side of the room but chose at random when they swapped positions. Thus, these findings might have been the consequence of local enhancement, however strong conclusions cannot be drawn for several reasons, which are discussed below.

To evaluate the occurrence of eavesdropping, the duration of looking, interacting, proximity and gaze alternations towards the experimenters and the first experimenter dogs looked at have been examined in previous studies [11,13,15,16,26]. Although a significant difference was found in the first experimenter dogs looked at in this study, there were no significant differences in any of the other behaviours towards the helpful and unhelpful experimenter. Furthermore, the impossible task was one out of two tests we conducted—in the choice test, dogs did not significantly choose the helpful person over the unhelpful person, regardless of whether the experimenters stayed on the same side or swapped positions.

We used the two-person impossible task, which is similar to Piotti et al.'s [26] first experiment. They used this setup to test whether dogs would look at the skilful experimenter over

the unskilful one after observing them manipulate a problem-solving apparatus. They found no significant difference in dogs' looking behaviour between the two experimenters and suggested that dogs may have evolved a strong predisposition to request human help regardless of their previous behaviour, as one domestication hypothesis [35,36] states that dogs adapted to life with humans and specialized in their capacity to communicate with humans, especially in cooperative contexts [37,38]. However, the looking back behaviour in the impossible task may be over-interpreted and does not represent a social 'help-seeking' strategy [39,40]–looking at the human in the impossible task is directly linked to the amount of time the animals spend interacting with the apparatus, thus it seems to be a by-product of 'giving-up' [39]. Additionally, a recent study by Lazzaroni, Marshall-Pescini, Manzenreiter, Gosch, Přibilová, Darc, McGetrick and Range [41] found that dogs look at the most salient stimulus after giving up on manipulating the impossible apparatus, regardless of whether the stimuli is a human, a human-like cut-out figure or a cardboard box. That being said, the two-person setup may be a better way to measure looking back behaviour to request help because both experimenters were equally salient in this setup. In this study, the fact that dogs looked back at the person only when they stayed in the same location suggests that it is the salience of the location that is the important factor rather than the interaction they had observed with the people present.

Our study also aimed to improve previous methodologies. However, by controlling for so many factors, the current experimental setup may have been too difficult for the dogs to understand the contingencies. First, the experimenters' actions in the observation phase may have been too similar. Both experimenters performed a pushing action and the only difference was that the helpful person slid the lid open, whereas the unhelpful person pushed the box away. The dog was also 3.5 m away when they observed the interactions, which may have been too far for them to see the minor difference in the experimenters' actions and their consequences clearly. If the animals could not discriminate between the helpful and unhelpful experimenters' actions, this may explain why we did not find a significant result in the choice test. This experiment could be altered to be more like Chijiiwa et al. [17], where subjects chose between the experimenter (who acted either helpfully or unhelpfully towards the owner) and a neutral person, who did not interact with the owner. As only one person interacted with the beggar in their study, this may have made it easier for dogs to discriminate between the two people.

Second, in the current study, the beggar also performed a similar action after interacting with both experimenters. After the helpful person opened the lid, the beggar ate the food, and after the unhelpful person pushed the box away, the beggar scratched her chin. This may have been problematic for two reasons: Marshall-Pescini, Passalacqua, Miletto Petrazzini, Valsecchi and Prato-Previde [42] found that the hand-to-mouth movement was a very salient cue for dogs. They investigated how nine different human ostensive cues influenced dogs' choice in a quantity discrimination task and found that dogs in the group that observed a human move her open and visibly empty hand from the plate of food to her mouth paid the most attention to the demonstration. This may explain why previous studies [15,16] found positive results, as the hand-to-mouth movement was performed exclusively on the giving donor's side. In contrast, the beggar stood equidistant to both experimenters when she performed the hand-to-mouth movement in the present study. Therefore, although our setup was an improvement on previous studies because we controlled for stimulus and local enhancement, it is possible that dogs were unable to discriminate between the beggar's different actions. The second reason is that the mere presence of food may explain the non-significant results because dogs may have paid more attention to the food than the behaviour of the experimenters. This idea is supported by Nitzschner et al. [13], who reported that dogs did not develop a preference for the giving donor even after many direct experiences in a pilot phase because they were too focused on the food. Chijiiwa et al. [17] used a neutral object (a roll of vinyl tape) instead and found

that dogs were biased against the unhelpful person. This is surprising, as a roll of vinyl tape has no relevance to dogs, but it does suggest that the dogs paid attention to the actor's behaviour. Thus, it might have been better to use a neutral object instead of food in this study.

Another explanation for the non-significant results may be related to the familiarity of the people in the test. When a stranger played the role of the beggar in Nitzschner et al.'s [16] first experiment, they found that dogs did not show a preference for the generous person. However, they found an effect when the owner played the role of the beggar in their second experiment. Chijiiwa et al. [17] also used owner-experimenter interactions and found that dogs avoided a person who behaved negatively towards their owner. These results indicate that the owner might have enhanced the salience of the experimenters' roles and so dogs may have paid more attention to the interactions. This idea is supported by other research that found owner-stranger effects; for example, Elgier, Jakovcevic, Mustaca and Bentosela [43] evaluated dogs' learning effects on the use of the pointing gesture in an object choice task and found that extinction was slower but reversal learning was faster when the owner gave the cue than when a stranger did.

A possible confound of this study is that dogs may not have understood why the beggar was requesting help in the observation phase. We chose this scenario because it was similar to what the dogs would then be presented with in the test phase, i.e., the impossibility of opening a box and the potential for choosing between two experimenters from whom to request for help. Although the interactions between the two humans in the observation phase may not have been very relevant to dogs, we tried to make them as naturalistic as possible. We included verbal communication, since it increased dogs' attentiveness towards the interactions, and non-verbal communication, which was likely more comprehensible to dogs. Such a scenario is arguably at least as naturalistic as others previously used, which appeared to show eavesdropping in dogs, for example, a human kneeling and silently begging for food like a dog [15, 16]. Nevertheless, it is still possible that dogs would have been more attentive in a more ecologically valid situation, such as humans interacting with another dog. For example, Rooney and Bradshaw [12] found that dogs preferred a human who won a tug-of-war game with another dog and suggested that winners of games are perceived as desirable social partners. Subiaul et al. [6] also found evidence for eavesdropping when chimpanzees observed third-party interactions between a humans and a conspecific and not between two humans. Therefore, it would be interesting to conduct a similar study to Marshall-Pescini et al. [15] and replace the human beggar with a conspecific to test whether the subject would choose to beg from a generous or a selfish human after observing them interact with another dog.

In conclusion, our study does not provide support for dogs being capable of eavesdropping. Instead, it provides weak evidence to support a local enhancement interpretation of eavesdropping, as dogs only preferred to look at the helpful person first in the impossible task if they stayed on the same side of the room before the animal's choice. A strong local enhancement interpretation would have been supported by animals preferring the unhelpful person after the experimenters had swapped positions, which was not the case in our study. However, since the dogs also did not show clear patterns of preferring the helpful person even if they did not swap positions, it is unclear whether they understood the current setup and the different actions of the experimenters. Increasing the relevance and ecological validity of the situation for the dogs, such as observing dog-human interactions, may help increase the dogs' attention to the setup and hence answer the question regarding their ability to eavesdrop.

## Supporting information

**S1 Table. Individual characteristics of dogs.**
(DOCX)

**S2 Table. Results of the full models in the impossible task.** Estimates, standard error, confidence intervals, results of significance tests and minimum and maximum of model estimates derived after excluding individuals one at a time.
(DOCX)

**S1 Video. Example of the procedure for the experimental condition.** The video shows the observation phase and the test phase for the impossible task and the choice test, which were counterbalanced across subjects. The video of the impossible task shows the footage from the experimenters' GoPros, which was synchronized and merged into one video.
(MP4)

**S2 Video. Example of the procedure for the side control condition.** The video shows the observation phase and the test phase for the choice test and the impossible task, which were counterbalanced across subjects. The video of the impossible task shows the footage from the experimenters' GoPros, which was synchronized and merged into one video.
(MP4)

**S1 Dataset. This dataset shows which experimenter the dogs looked at first in the impossible task and which experimenter they approached in the choice test.**
(CSV)

**S2 Dataset. This dataset shows the duration (0.2 s) of looking, interacting, and proximity to the helpful and unhelpful experimenters, and the frequency of 2-way and 3-way gaze alternations combined towards each experimenter in the impossible task.**
(CSV)

## Acknowledgments

We thank the Clever Dog Lab (University of Veterinary Medicine Vienna), which provided the contacts of the dog owners and the facilities to conduct the study. We thank the dog owners for participating and Simone Bruckner, Alina Gaugg, Eléonore Rolland, Anja Uellendahl, Roberta Vitiello and Julie Welter for acting as the experimenters in the study. We also thank Riccardo Hofer and Jinook Oh for the technical support and Nadja Kavcik-Graumann for help with Fig 3. Finally, we thank Roger Mundry for the statistical advice.

## Author Contributions

**Conceptualization:** Hoi-Lam Jim, Sarah Marshall-Pescini, Friederike Range.

**Data curation:** Hoi-Lam Jim.

**Formal analysis:** Hoi-Lam Jim.

**Funding acquisition:** Friederike Range.

**Investigation:** Hoi-Lam Jim, Sarah Marshall-Pescini, Friederike Range.

**Methodology:** Hoi-Lam Jim, Sarah Marshall-Pescini, Friederike Range.

**Project administration:** Friederike Range.

**Resources:** Friederike Range.

**Supervision:** Sarah Marshall-Pescini, Friederike Range.

**Writing – original draft:** Hoi-Lam Jim.

**Writing – review & editing:** Hoi-Lam Jim, Sarah Marshall-Pescini, Friederike Range.

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
