## [Decision Letter · Decision Letter 0]

28 Apr 2020

PONE-D-20-00176

Do dogs eavesdrop on human interactions in a helping situation?

PLOS ONE

Dear Miss Jim,

Thank you for submitting your manuscript to PLOS ONE. After careful consideration, we feel that it has merit but does not fully meet PLOS ONE’s publication criteria as it currently stands. Therefore, we invite you to submit a revised version of the manuscript that addresses the points raised during the review process.

As noted by all of the reviewers, there should be clarification of some terms, details of the subjects, experimental procedure. In addition, there are suggested changes to some sentences based on grammatical points. All of the reviewers also note that the findings are not particularly conclusive, but you have done a thorough job of addressing concerns and possible issues with the experiment in the discussion. One of the reviewers saw this as a major flaw and suggests that you replicate the study addressing some of the concerns; however, two of the reviewers note that the paper is still publishable as is. My suggestion is to incorporate all suggestions and changes to the current manuscript. If you have any data you can add (or reference an additional study or experiment that has been conducted since the original study), please include that in the revision. 

We would appreciate receiving your revised manuscript by Jun 12 2020 11:59PM. To enhance the reproducibility of your results, we recommend that if applicable you deposit your laboratory protocols in protocols.io, where a protocol can be assigned its own identifier (DOI) such that it can be cited independently in the future. For instructions see: http://journals.plos.org/plosone/s/submission-guidelines#loc-laboratory-protocols

We look forward to receiving your revised manuscript.

Kind regards,

Julie Jeannette Gros-Louis, PhD

Academic Editor

PLOS ONE

Reviewers' comments:

Reviewer's Responses to Questions

**Comments to the Author**

1. Is the manuscript technically sound, and do the data support the conclusions?

Reviewer #1: Yes

Reviewer #2: No

Reviewer #3: Yes

2. Has the statistical analysis been performed appropriately and rigorously? 

Reviewer #1: Yes

Reviewer #2: Yes

Reviewer #3: Yes

3. Have the authors made all data underlying the findings in their manuscript fully available?

Reviewer #1: Yes

Reviewer #2: Yes

Reviewer #3: Yes

4. Is the manuscript presented in an intelligible fashion and written in standard English?

Reviewer #1: Yes

Reviewer #2: Yes

Reviewer #3: Yes

5. Review Comments to the Author

Reviewer #1: This is a well-written paper and the experiment and analyses appear to be rigorously conducted. The results seem fairly clear and the discussion is thorough and balanced. I have some relatively minor comments:

There is a clear rationale for the main aspect of the study. The topic of eavesdropping/reputation is extremely interesting and important and somewhat understudied across species. It is nice to see authors delving into the mechanisms responsible for the mixed results in dogs. However, the authors mention the impossible task as somewhat of an afterthought at the end of the introduction. The rationale for this part of the study should be better incorporated into the rest of the background.

They might include more of a background for why dogs should have evolved the ability to form reputations (if they believe this).

In Subiaul et al., it could not be ruled out that the chimpanzees learned to attend to the giving experimenter rather than forming a reputation of “generous.”

The task is somewhat artificial. Dogs typically do not open boxes and thus may find it difficult to reason about how and why the experimenter is requesting help. Could the authors address the artificiality of the paradigm? It seems a bit silly to have the experimenters communicate verbally. Wouldn’t it be better to have a nonverbal exchange given the dogs can’t understand human speech? In the figure, the actions look very similar as if only the language differentiates helpful from unhelpful.

There is at least one study on reputation/eavesdropping in domestic cats that produced a null result. However, it was based on a more naturalistic examination of how cats responded to friendly and aggressive strangers attending only to natural behaviors. It should be available soon but is currently in press here:

Leete, J.A., Vonk, J., Oriani, S., Eaton, T., & Lieb, J. (2020). Domestic cats (Felis silvestris catus) do not infer reputation in humans after direct and indirect experience. Human-Animal Interaction Bulletin, 8.

The authors should specify that dogs were included on the basis of lack of experience with the experimenters.

Use a , after i.e. and e.g.

Watch the placement of “only.”

The authors should provide detail of the dogs’ breeds, sex and ages in a table.

Was the order of tests counterbalanced within the experimental conditions or just in general?

I am glad that the authors address that looking at the experimenter may simply be related to the amount of time spent working on the problem in the impossible task.

Some of the figures are unnecessary (e.g. Fig 4).

Reviewer #2: The paper is interesting and studies a relevant and controversial topic in the area. However, even when the aim of the authors was to increase the control over several variables, in my opinion the procedure was inadequate. As the authors clearly stated in the discussion, the task was too difficult and the differences between the helpful and unhelpful experimenters were minimal. In addition the consequences of the dogs´actions were not contingent with their behavior during the first test, and this could affect the responses of the dogs during the second test. This situation could be frustrating for the dogs and it might extinguish any preference for one experimenter.

On the other hand, if I understood correctly, the person did not show that she was unable to open the box. Therefore, it is not clear if it was a helping situation according to the dogs´perspective. Maybe dogs could only perceive different reactions of each experimenter toward the box.

In conclusion, there are no clear indications that dogs have discriminated between the behavior of the two experimenters, even in the condition in which they stayed in the same place. Unfortunately, considering that you could not replicate the basic phenomenon, no valid conclusions could be obtained from this study. In my opinion, the manuscript can not be accepted in the present form. However, I strongly encourage the authors to replicate the study keeping the control of the variables, but changing the procedure in order to increase the differences between the two experimenters.

Other details to consider

You have to include the data of the characteristics (breed, age, sex) of the subjects of each group. Also, you must clarify if dogs had been evaluated in other tasks.

L 160 Were the dogs unleash?

“The positions were marked by tape on the floor”: include this description also in the experimental setup section.

L 170 How long was the habituation period?

L 177 How did the owner hold the dog? By its collar?

I suggest including a video of the procedure

L 229 “The owner was allowed to give a short prompt if the dog did not move”. I do not understand this. If the owners were blindfolded, how did they prompt dogs? Could they influence with this procedure the dogs behavior?

It would be interesting to analyze the effects of sex and age on dogs behavior.

Reviewer #3: General comments:

This paper is well thought out and thorough. The finding is not particularly conclusive but the study was conducted rigorously and the data is sound. It is particularly noteworthy that the study in the paper contradicts previous implications of the authors earlier work.

Where I saw confounds, they were well addressed in the discussion, notably unfamiliar person effects, food salience effects, and the highly nuanced distinction between helping and hindering behaviors. The design eliminated the confounds it was intending to, but due to the complexities of the new method (e.g., the hand to mouth action of the experimenter was potentially too subtle for dogs to differentiate between conditions), this study was limited in the amount of clarity it adds to the original study published previously. Although a followup study would be necessary to draw strong conclusions, the discussion presents a clear understanding of where this piece of research would fall within the broader question.

There are several instances of run-on, or grammatically opaque sentences that could be clearer if phrased differently. Notable examples include sentences beginning on lines 57, 66, 121, and 147. In order for me to recommend this paper for publication, I would need the authors to conduct a more thorough editing job to ensure the clarity of these longer sentences (e.g., rewording them or dividing them into shorter sentences.

Introduction

Line 93 - The word “choice” is used as a dependent variable before explaining what kind of choice dogs are performing. It would be clearer to define the measure as soon as it is introduced.

Method

What is a bum bag? It seems that it might be what we call a fanny pack, but the terminology should be looked into for clarity across the widest possible audience.

Line 177 - It is not clear from this description exactly where the food is placed. The wording could be more precise.

The control condition introduced involves having the experimenter switch sides. This could be relabeled Side Control condition to make it clearer that having the experimenter switch sides is to control for a side bias.

Line 217 - How is the apparatus closed between trials? It mentions that six trials are run in a row, but if the beggar is requesting help in each trial, how is the apparatus rendered unavailable to the beggar between trials?

For dogs that made no choices and did not receive food from either experimenter, were there any effects of trial number on their performance?

Line 264 - The text should refer to table in text for the coding definition.

Table 1 - 6 - Does the dog have to be tail wagging and in close contact, or just one of the two, in order for this behavior to be coded?

Discussion

I had two major concerns about the experimental design. One was that the design might be so complicated that the authors might have made the demonstration too confusing for the dog to understand the necessary information. The second was that the controls might have left too little to differentiate the actions, especially with very salient clues like bringing a hand toward the mouth. However, both of these concerns were thoroughly addressed in the discussion.

6. PLOS authors have the option to publish the peer review history of their article (what does this mean?). If published, this will include your full peer review and any attached files.

Reviewer #1: No

Reviewer #2: No

Reviewer #3: No

---

## [Author Response · Author response to Decision Letter 0]

18 Jul 2020

Response to Reviewer #1:

This is a well-written paper and the experiment and analyses appear to be rigorously conducted. The results seem fairly clear and the discussion is thorough and balanced. I have some relatively minor comments:

There is a clear rationale for the main aspect of the study. The topic of eavesdropping/reputation is extremely interesting and important and somewhat understudied across species. It is nice to see authors delving into the mechanisms responsible for the mixed results in dogs. However, the authors mention the impossible task as somewhat of an afterthought at the end of the introduction. The rationale for this part of the study should be better incorporated into the rest of the background.

We have made this change (line #97):

Chijiiwa et al. [17] is the only study that has tested eavesdropping in dogs in a helping situation, and every study to date on this topic has measured whether dogs attributed reputations to humans based on the dog’s first approach or time spent in close proximity to each experimenter. Another way to measure dogs’ preference for a human in a helping situation is the ‘impossible task’ [18]. This a well-established paradigm where the dog is presented with an apparatus containing food that cannot be accessed. This usually elicits looking behaviour at a human, which has been interpreted as a communicative act to request help from humans [19–26]. When two people are present during this task, it can be used to test whether dogs can discriminate between people – for example, Marshall-Pescini, Passalacqua, Barnard, Valsecchi and Prato-Previde [23] found differences in dogs’ looking behaviour towards their owner and the researcher based on their life experiences, whereas Piotti, Spooner, Jim and Kaminski [26] found no significant difference in their looking behaviour between a skilful human and an unskilful one. We used a variation of the impossible task that was similar to Piotti et al.’s [26] first experiment to test whether dogs would look at a helpful person first or for significantly longer over an unhelpful person after observing them acting helpfully or unhelpfully towards another person, possibly to request help to open the box to retrieve the food inside.

They might include more of a background for why dogs should have evolved the ability to form reputations (if they believe this).

We have added our rationale for why we think dogs should have evolved the ability to form reputations (line #52):

A problem with studying eavesdropping in non-human apes is that the situation is highly artificial, as they do not typically rely on humans for social information. Therefore, studying animals that live and interact with humans, i.e., domesticated species, would be more ecologically valid. A recent study by Leete, Vonk, Oriani, Eaten and Lieb [9] showed that domestic cats (Felis silvestris catus) did not attribute reputations to humans through direct or indirect interactions between humans and conspecifics. A possible explanation for this result is that cats are typically solitary animals and do not rely on cooperation with others for survival. Conversely, dogs (Canis lupus familiaris) descend from wolves (Canis lupus), which live in family groups and engage in group hunting and alloparenting [10], thus they are highly social and cooperative. Furthermore, dogs were domesticated by humans – they can cooperate with humans and rely on them for valuable resources [11], such as food and shelter. Thus, it would be beneficial for dogs to observe humans interacting with each other or with other dogs and gain information from such an exchange to select the most appropriate person with whom to associate and whom to avoid. 

In Subiaul et al., it could not be ruled out that the chimpanzees learned to attend to the giving experimenter rather than forming a reputation of “generous.”

Thank you for pointing this out. We have made this change (line #46):

Subiaul, Vonk, Okamoto-Barth and Barth [6] also tested chimpanzees in the same setup and did not find any significant results. They conducted a follow-up experiment where a conspecific was the recipient of the interaction and found that chimpanzees preferred the nice person over the nasty person. However, it could not be ruled out that the chimpanzees learned to attend to the nice experimenter rather than forming a reputation of “generous” in this study.

The task is somewhat artificial. Dogs typically do not open boxes and thus may find it difficult to reason about how and why the experimenter is requesting help. Could the authors address the artificiality of the paradigm? It seems a bit silly to have the experimenters communicate verbally. Wouldn’t it be better to have a nonverbal exchange given the dogs can’t understand human speech? In the figure, the actions look very similar as if only the language differentiates helpful from unhelpful.

We have addressed this in the discussion (line #496):

A possible confound of this study is that dogs may not have understood why the beggar was requesting help in the observation phase. We chose this scenario because it was similar to what the dogs would then be presented with in the test phase, i.e., the impossibility of opening a box and the potential for choosing between two experimenters from whom to request for help. Although the interactions between the two humans in the observation phase may not have been very relevant to dogs, we tried to make them as naturalistic as possible. We included verbal communication, since it increased dogs’ attentiveness towards the interactions, and non-verbal communication, which was likely more comprehensible to dogs. Such a scenario is arguably at least as naturalistic as others previously used, which appeared to show eavesdropping in dogs, for example, a human kneeling and silently begging for food like a dog [15, 16].

There is at least one study on reputation/eavesdropping in domestic cats that produced a null result. However, it was based on a more naturalistic examination of how cats responded to friendly and aggressive strangers attending only to natural behaviors. It should be available soon but is currently in press here:

Leete, J.A., Vonk, J., Oriani, S., Eaton, T., & Lieb, J. (2020). Domestic cats (Felis silvestris catus) do not infer reputation in humans after direct and indirect experience. Human-Animal Interaction Bulletin, 8.

Thank you for informing us about this new interesting study. We have integrated this into the manuscript by explaining how studying eavesdropping in non-human apes using human interactions is not as ecologically valid as for domesticated species, such as cats and dogs (line #52).

The authors should specify that dogs were included on the basis of lack of experience with the experimenters.

We have made this change (line #149):

They were selected from a database of owners who had volunteered to participate in behavioural studies at the Clever Dog Lab and were included on the basis of lack of experience with the experimenters in this study.

This was also mentioned in the Experimental setup section (line #180):

There were three experimenters: the main experimenter, who acted as the beggar and set-up the trials, and two female experimenters who were unfamiliar to the dog.

Use a , after i.e. and e.g.

Thank you for pointing this out. We have made this change throughout the manuscript.

Watch the placement of “only.”

Thank you for pointing this out. As a native speaker, I have gone through the manuscript and made the appropriate changes. My co-authors have also read through it and we hope that it is fine now.

The authors should provide detail of the dogs’ breeds, sex and ages in a table.

We have added a table of the dogs’ characteristics in the Supporting information (S1 Table).

Was the order of tests counterbalanced within the experimental conditions or just in general?

The order of tests was counterbalanced within the experimental conditions – we have made this clearer in the methods section (line #224):

Which of the two tests was conducted first (impossible task or choice test) was counterbalanced across subjects within the experimental groups.

I am glad that the authors address that looking at the experimenter may simply be related to the amount of time spent working on the problem in the impossible task.

Some of the figures are unnecessary (e.g. Fig 4).

We have removed Figure 4 and provided videos of the procedure in the Supporting information instead (S1 Video and S2 Video).

 

Response to Reviewer #2:

The paper is interesting and studies a relevant and controversial topic in the area. However, even when the aim of the authors was to increase the control over several variables, in my opinion the procedure was inadequate. As the authors clearly stated in the discussion, the task was too difficult and the differences between the helpful and unhelpful experimenters were minimal. In addition the consequences of the dogs´ actions were not contingent with their behavior during the first test, and this could affect the responses of the dogs during the second test. This situation could be frustrating for the dogs and it might extinguish any preference for one experimenter.

We acknowledge the complexity of the task and included this in the discussion, however previous studies with paradigms of arguably similar complexity did find an ‘eavesdropping effect’. To account for the possibility that the dogs’ actions in the first test affected their behaviour in the second test, we included test order in the models as a predictor and it did not have a significant effect in any of the models (see Table 3 and S2 Table). Therefore, the dogs did not show any preference for one experimenter in the first test and it did not have an effect on their preference for the second test.

On the other hand, if I understood correctly, the person did not show that she was unable to open the box. Therefore, it is not clear if it was a helping situation according to the dogs´ perspective. Maybe dogs could only perceive different reactions of each experimenter toward the box.

The beggar did not show that she was unable to open the box, but humans do not always try to do something, such as open a box, before they ask someone for help. As we mentioned in the manuscript, we tried to make the interactions as naturalistic as possible, and it is plausible that dogs would observe such a situation in their everyday life. Therefore, if dogs do eavesdrop, we would have expected them to show eavesdropping in this scenario. We also found that the proportion of dogs that looked at the helpful experimenter first in the impossible task was significantly above chance if the experimenter did not swap sides, which suggests that the dogs were able to perceive the small differences in the experimenter’s actions.

In conclusion, there are no clear indications that dogs have discriminated between the behavior of the two experimenters, even in the condition in which they stayed in the same place. 

We found that the proportion of dogs that looked at the helpful experimenter first in the impossible task was significantly above chance when the experimenters stayed in the same position (p = .033), which suggests that the dogs were able to discriminate between the two experimenters. However, this does not represent eavesdropping, as the dogs may have simply gone to the location of a positive action (the helpful interaction) rather than paid attention to who was doing what. Thus, the rationale for our study was to test the local enhancement hypothesis, so the experimenters swapped positions for half of the sample. However, it was more challenging for the dogs to keep track of the experimenters in the side control group.

Unfortunately, considering that you could not replicate the basic phenomenon, no valid conclusions could be obtained from this study. In my opinion, the manuscript cannot be accepted in the present form. However, I strongly encourage the authors to replicate the study keeping the control of the variables, but changing the procedure in order to increase the differences between the two experimenters.

The eavesdropping phenomenon in dogs is at present quite controversial, as outlined by the mixed results from previous studies. There is therefore a need to publish both positive and negative results to gain a better understanding of the features which may affect the occurrence of such a phenomenon in dogs. 

Other details to consider

You have to include the data of the characteristics (breed, age, sex) of the subjects of each group. Also, you must clarify if dogs had been evaluated in other tasks.

We have added a table of the dogs’ characteristics and whether they had been evaluated in other tasks in the Supporting information (S1 Table). We also mentioned it in the Subjects section (line #151):

25 dogs had participated in other studies.

L 160 Were the dogs unleash?

Yes – we have clarified this point in the manuscript (line #211): 

Once the dog was comfortable, the owner sat and held his/her unleashed dog by the collar between his/her legs while the main experimenter placed a piece of sausage in the clear box in the centre of the impossible task apparatus. 

“The positions were marked by tape on the floor”: include this description also in the experimental setup section.

We have clarified this in the Experimental Setup section (line #188):

Before the trial with the impossible task began, the apparatus was placed in a predetermined position in the centre of the room, 1.5 m away from the dog (Fig 2). During the observation phase, the beggar stood in position B, 3.5 m away from the dog. The two experimenters stood 1.5 m away on either side of the beggar in position A and C. During the test phase, the experimenters walked 2 m forward and stood in position D and E. Two semi-circles with a radius of 1 m from positions D and E facing towards each other were marked by tape on the floor to indicate when the dog was in close proximity to the experimenter.

L 170 How long was the habituation period?

We have clarified this in the manuscript (line #209):

Prior to testing, the subject was allowed to explore the room and the impossible task apparatus freely for approximately 5 minutes while the main experimenter explained the procedure to the owner. 

L 177 How did the owner hold the dog? By its collar?

Yes – we have clarified this in the manuscript (line #211). 

I suggest including a video of the procedure

Thank you for the suggestion. We have provided two videos in the Supporting information: S1 Video shows the procedure for the experimental condition and S2 Video shows the procedure for the side control condition. Each video shows the observation phase and the test phase for the impossible task and the choice test, which were counterbalanced. The video of the impossible task shows the footage from the experimenters’ GoPros, which was synchronized and merged into one video and used for coding.

L 229 “The owner was allowed to give a short prompt if the dog did not move”. I do not understand this. If the owners were blindfolded, how 

did they prompt dogs? Could they influence with this procedure the dogs behavior?

We have clarified this in the manuscript (line #267): 

The owner was allowed to give a short prompt if the dog did not move by itself, such as a gentle nudge or saying “ok” to indicate to the dog that they were free to move, but the owner was instructed not to gesture in a specific direction. The owners were blindfolded so they were not aware of the experimenters’ positions and could not influence their dog’s behaviour.

It would be interesting to analyze the effects of sex and age on dogs behavior.

We did not hypothesize about effects of sex and age, thus we did not analyze it.

 

Response to Reviewer #3:

General comments:

This paper is well thought out and thorough. The finding is not particularly conclusive but the study was conducted rigorously and the data is sound. It is particularly noteworthy that the study in the paper contradicts previous implications of the authors earlier work.

Where I saw confounds, they were well addressed in the discussion, notably unfamiliar person effects, food salience effects, and the highly nuanced distinction between helping and hindering behaviors. The design eliminated the confounds it was intending to, but due to the complexities of the new method (e.g., the hand to mouth action of the experimenter was potentially too subtle for dogs to differentiate between conditions), this study was limited in the amount of clarity it adds to the original study published previously. Although a followup study would be necessary to draw strong conclusions, the discussion presents a clear understanding of where this piece of research would fall within the broader question.

There are several instances of run-on, or grammatically opaque sentences that could be clearer if phrased differently. Notable examples include sentences beginning on lines 57, 66, 121, and 147. In order for me to recommend this paper for publication, I would need the authors to conduct a more thorough editing job to ensure the clarity of these longer sentences (e.g., rewording them or dividing them into shorter sentences.

Thank you for pointing this out. As a native speaker, I have gone through the manuscript and made the appropriate changes. My co-authors have also read through it and we hope that it is fine now.

Introduction

Line 93 - The word “choice” is used as a dependent variable before explaining what kind of choice dogs are performing. It would be clearer to define the measure as soon as it is introduced.

We have clarified this in line #97:

Chijiiwa et al. [17] is the only study that has tested eavesdropping in dogs in a helping situation, and every study to date on this topic has measured whether dogs attributed reputations to humans based on the dog’s first approach or time spent in close proximity to each experimenter.

and in line #122:

Second, like previous studies, we measured dogs’ preference considering their choice to approach an experimenter (helpful or unhelpful) after witnessing the third-party interactions.

Method

What is a bum bag? It seems that it might be what we call a fanny pack, but the terminology should be looked into for clarity across the widest possible audience.

Thank you for pointing this out. We have changed it to ‘hip bag’ throughout the manuscript.

Line 177 - It is not clear from this description exactly where the food is placed. The wording could be more precise.

We have clarified this in line #211:

Once the dog was comfortable, the owner sat and held his/her unleashed dog by the collar between his/her legs while the main experimenter placed a piece of sausage in the clear box in the centre of the impossible task apparatus.

The control condition introduced involves having the experimenter switch sides. This could be relabeled Side Control condition to make it clearer that having the experimenter switch sides is to control for a side bias.

Thank you for this suggestion. We have made this change in the Experimental design section (line #202):

Dogs were randomly assigned to two experimental groups: experimental and side control. In the experimental group, the experimenters stayed on the same side of the room in the test phase. In the side control group, the experimenters swapped positions in the test phase to test the local enhancement hypothesis. Half of the sample was in the experimental group and the other half was in the side control group.

Line 217 - How is the apparatus closed between trials? It mentions that six trials are run in a row, but if the beggar is requesting help in each trial, how is the apparatus rendered unavailable to the beggar between trials?

We have clarified this in line #245 and suggest watching the videos in the Supporting information:

After the response, the beggar returned to her position in-between the experimenters and faced the dog. In the instance of having interacted with the helpful person, she took a piece of sausage out of the box, showed it to the dog then put it in her mouth and chewed (i.e., moved her jaw up and down) to make it clear to the dog that she was eating. At the same time, she moved the box towards her chest to close the lid inconspicuously, so that it was closed for the next interaction. In the instance of having interacted with the unhelpful person, the beggar scratched her chin whilst the box remained closed. Thus, the beggar’s actions were similar after interacting with both experimenters to ensure the dog did not prefer the helpful person because the beggar performed a particular action after interacting with her (see videos in Supporting information).

For dogs that made no choices and did not receive food from either experimenter, were there any effects of trial number on their performance?

The seven dogs that did not make a choice or receive food from either experimenter in the choice test were excluded from the analysis because they did not perform a behaviour to indicate a preference for an experimenter.

Line 264 - The text should refer to table in text for the coding definition.

We have made this change (line #293):

We coded the following behaviours at 0.2 s time resolution: the frequency of (1) 2-way gaze alternations and (2) 3-way gaze alternations and the duration of (3) looking at the box; (4) interacting with the box; (5) looking at the experimenter; (6) interacting with the experimenter; (7) proximity to the experimenter (see Table 1 for definitions).

Table 1 - 6 - Does the dog have to be tail wagging and in close contact, or just one of the two, in order for this behavior to be coded?

We have clarified this in Table 1 (6):

The dog being in close contact and exhibiting social behaviours towards the experimenter, e.g., sniffing and tail-wagging whilst in close contact

Discussion

I had two major concerns about the experimental design. One was that the design might be so complicated that the authors might have made the demonstration too confusing for the dog to understand the necessary information. The second was that the controls might have left too little to differentiate the actions, especially with very salient clues like bringing a hand toward the mouth. However, both of these concerns were thoroughly addressed in the discussion.

Additional changes:

We have added a sentence about the individuals consenting to the videos in the Supplementary information in line #164:

The individuals in this manuscript have given written informed consent (as outlined in PLOS consent form) to publish the videos in the Supplementary information.

We obtained permission from Nadja Kavcik-Graumann for her drawing in Figure 3 and we have attached it as an “Other” file.

We have added the two datasets analyzed in the Supplementary information.

---

## [Editor Report · Decision Letter 1]

27 Jul 2020

Do dogs eavesdrop on human interactions in a helping situation?

PONE-D-20-00176R1

Dear Dr. Jim,

We’re pleased to inform you that your manuscript has been judged scientifically suitable for publication and will be formally accepted for publication once it meets all outstanding technical requirements.

Kind regards,

Julie Jeannette Gros-Louis, PhD

Academic Editor

PLOS ONE
---

## [Editor Report · Acceptance letter]

31 Jul 2020

PONE-D-20-00176R1 

Do dogs eavesdrop on human interactions in a helping situation? 

Dear Dr. Jim:

I'm pleased to inform you that your manuscript has been deemed suitable for publication in PLOS ONE. Congratulations! Your manuscript is now with our production department. 

Kind regards, 

on behalf of

Dr. Julie Jeannette Gros-Louis 

Academic Editor

PLOS ONE